# Antibacterial Effects of X-ray and MRI Contrast Media: An In Vitro Pilot Study

**DOI:** 10.3390/ijms24043470

**Published:** 2023-02-09

**Authors:** Michael P. Brönnimann, Lea Hirzberger, Peter M. Keller, Monika Gsell-Albert

**Affiliations:** 1Department of Diagnostic, Interventional and Paediatric Radiology, Inselspital, Bern University Hospital, University of Bern, 3010 Bern, Switzerland; 2Institute for Infectious Diseases, University of Bern, 3010 Bern, Switzerland

**Keywords:** contrast media, anti-bacterial agents, microbial sensitivity tests, antibacterial effect, magnetic resonance imaging, interventional, minimally invasive surgical procedures, anti-infective agents

## Abstract

Some radiological contrast agents have been shown to have effects on bacterial growth. In this study, the antibacterial effect and mechanism of action of iodinated X-ray contrast agents (Ultravist 370, Iopamiro 300, Telebrix Gastro 300 and Visipaque) and complexed lanthanide MRI contrast solutions (MultiHance and Dotarem) were tested against six different microorganisms. Bacteria with high and low concentrations were exposed to media containing different contrast media for various lengths of time and at pH 7.0 and 5.5. The antibacterial effect of the media was examined in further tests using agar disk diffusion analysis and the microdilution inhibition method. Bactericidal effects were found for microorganisms at low concentrations and low pH. Reductions were confirmed for *Staphylococcus aureus* and *Escherichia coli*.

## 1. Introduction

CT- or MR-guided interventions often require luminal administration of contrast media for diagnostic purposes such as characteristics of the abscess cavity, positional control of drainage, or detection of a fistula tract to adjacent structures. The main components are the chemically bound iodine in X-ray contrast media and gadolinium in MRI contrast media. First, in 1839, John Davis, a surgeon, used a tincture of iodine on the battlefield as an antiseptic and disinfectant [1]. The high toxicity of free lanthanides in the body, such as gadolinium, has been known since the middle of the last century [2]. As the development of contrast agents has progressed, every effort has been made to create stable compounds with these cytotoxic elements bound therein. The gadolinium ion (Gd^3+^) is highly toxic in its free form. Therefore, it is chelated with an organic ligand molecule to produce highly soluble nontoxic complexes [3].

Nevertheless, small amounts of these components (iodine, gadolinium) are known to dissociate freely [1,2]. The extent to which presumed free components have a bacteriostatic or bactericidal effect has been investigated little and stretches selectively over the last century. Different results with older, ionic X-ray contrast agents and common bacteriostatic effects of newer, nonionic X-ray contrast agents, primarily against Gram-negative bacteria, have been reported [4,5,6,7]. The data situation regarding MRI contrast agents is even thinner, with Green, Moustache [8] demonstrating a lack of proliferation of organisms and Beussink et al. [9] showing antimicrobial activity in tested MRI contrast agents. This in vitro pilot study aims to evaluate the antibacterial effect of commonly used X-ray and MRI contrast media from different chemical classification groups, considering pH, incubation time and carbon substrate utilization. Such diversification in contrast agents and other laboratory parameters has never been studied. We hypothesize that bacteriostatic effects can be shown.

## 2. Results

### 2.1. Antibacterial Efficacy Tests in Blood Culture Bottles

Blood culture bottles were inoculated with bacterial suspensions of the microorganisms *Staphylococcus aureus, Pseudomonas aeruginosa* and *Bacillus subtilis* in two different densities that were previously incubated with contrast agents Ultravist, Iopamiro, Telebrix Gastro or Visipaque. The time until growth of pathogens was recorded (Table 1). Negative controls with sodium chloride solution were integrated.

For the pathogen *S. aureus*, all blood culture bottles showed growth after a few hours of incubation (3.7 and 9.9 h for the high bacterial count and between 11.1 and 16.5 for the low bacterial count) with all X-ray contrast media. Incubation with X-ray contrast media for 48 h resulted in growth always being delayed compared to incubation for 24 h. Similarly, the blood culture bottles with the lower bacterial counts (5 mL of a suspension with 10^2^ CFU/mL was placed in 10 mL of blood culture liquid) always became positive later. The negative controls showed no growth anymore after 48 h incubation.

A similar picture was seen for the pathogen *B. subtilis*. However, with the X-ray contrast medium Ultravist 370, no growth was seen after 24 h at the low pH of 5.5 with the low bacterial count, but after 48 h, growth was seen again (Figure 1). The negative controls showed no growth after 48 h.

The results were similar for *P. aeruginosa*. Only the contrast medium Teblebrix Gastro 300 showed no growth at both pH values with the low bacterial count after 24 h; likewise, after 48 h at pH 7.1 (Figure 2). The negative controls with the low pH showed no more growth after 48 h.

### 2.2. Agar Disk Diffusion

Filter paper disks soaked in X-ray contrast medium were placed on plates inoculated with *B. subtilis*. The X-ray contrast agent’s inhibitory or bactericidal effect could be shown due to the inhibition zone that formed around the filter paper disks. Erythromycin-soaked filter paper disks were used as controls. The inhibition zone that formed around them had a diameter between 27 and 38 mm. At pH 7.4, only Ultravist 370 showed a minimal inhibition zone, whereas, at pH 5.5, small inhibition zones were detected for the contrast agents Ultravist 370, Telebrix Gastro 300, Multihance and Dotarem. In a second test series, fresh suspensions of *S. aureus* and *P. aeruginosa* were prepared and plated onto agar plates. Filter paper disks soaked with X-ray contrast medium were placed onto these plates. The X-ray contrast agent’s inhibitory or bactericidal effect could be shown due to the inhibition zone that formed around the filter paper disks. Erythromycin-soaked filter paper plates were used as controls. They had a diameter between 17 and 25 mm for the inhibition of *S. aureus*, whereas *P. aeruginosa* was not sensitive to erythromycin and showed no inhibition. All the contrast agents showed no inhibition either. They all had a diameter of 6 mm (corresponding to the diameter of the filter paper disk).

### 2.3. Broth Microdilution in Minimal Medium

Iodine and gadolinium ions are complexed with organic ligands and are, therefore, not available for bacteria. Thus, we tried to show that if the carbon source in the medium has been used up, the contrast agent itself can be used as a carbon source. By breaking up the contrast media, iodine or gadolinium ions are released and can have a toxic effect. For the broth microdilution experiment, microorganisms that have long doubling times were added. Dense, homogenous suspensions of *E.coli*, *S. aureus*, *M. smegmatis* and *F. necrophorum* were prepared. Afterwards, these cultures were transferred into minimal medium M9 supplemented with 2% glucose for culturing. A quantity of 1.5 mL was taken out and pelleted by centrifugation. The pellet was washed with PBS and transferred into minimal medium M9. A quantity of 100 µL of all bacterial strains in the two media were placed onto a microplate and incubated with 100 µL of each contrast agent. Growth controls in M9 supplemented with 2% glucose and M9 of all bacterial strains were included. After up to 48 h incubation, the growth was examined. After 48 h, the pathogen *M. smegmatis* showed growth with the following five X-ray contrast media; Ultravist, Iopamiro, Telebrix Gastro, Multihance and Dotarem (Table 2), Visipaque being the only contrast agent that has not shown growth. *S. aureus* has not grown in any of the X-ray contrast media. *E.coli* showed growth with the following X-ray contrast media: Iopamiro, Visipaque, Multihance (1/3) and Dotarem(1/3).

## 3. Discussion

The effect of representatives of all classification groups of iodinated X-ray contrast agents (monomeric, dimeric, high osmolar, low osmolar, ionic and nonionic) as well as gadolinium-based contrast media (macrocyclic vs. linear) on bacteria was investigated in our in vitro study by using three different tests. The most important findings are the lack of growth detection in broth microdilution in a minimal medium of *M. smegmatis* in Visipaque, of *E. coli* in Ultravist 370 and Telebrix Gastro, and of *S. aureus* in all tested contrast media. The additional lack of effect on *S. aureus* in the other tests (antibacterial efficacy test in the blood culture bottle and agar disk diffusion) supports our hypothesis that the components (free iodine and gadolinium) are consumed and a cytotoxic effect results. The lack of growth of negative controls with NaCl after 48 h in low suspension density also supports this. The bacteria probably died because of the lack of nutritious substances. The lack of a carbon source was not compensated. Still, we consider the controls to be adequate, since there was growth after 24 h. That the duration of contact between contrast media and microorganisms is an important factor was already assumed by Blake and Halasz [4].

This could also explain why in our results Multihance and Dotarem showed a lack of growth in *S. aureus* and *E. coli* in only one-third of our controls. Since our incubation period was only 86 h and not up to 28 days as in Beussink, Godat and Seaton [9]. Furthermore, this aspect is supported by our following vital results, such as a bacteriostatic effect on *B. subtilis* and the only bactericidal effect of Telebrix Gastro on *P. aeruginosa* (Figure 1 and Figure 2). On the one hand, we were able to reproduce the previously described bactericidal effect of X-ray contrast media, especially on Gram-negative bacteria [4,10,11]. Thus, we support the hypothesis that the different cell wall of gram-negative bacteria presumably is more strongly affected by the presence of X-ray contrast media [4]. On the other hand, this is presumably only true for the first 48 h of contact, as our other tests (agar disk diffusion and broth microdilution in minimal medium) could not substantiate this, and later, the released components (iodine or gadolinium) from the complex may cause the subsequent main effect. This is also supported by the fact that the free iodine content of ionic X-ray contrast media is up to 12 times higher than that of the more stable, nonionic X-ray contrast media [12]. Based on this, our inconsistent findings concerning ionic and nonionic X-ray contrast agents in the different tests can be explained and are in line with the previous literature [4]. This argument is also supported by the fact that even our bactericidal and bacteriostatic results were initially demonstrated only at low organism concentrations (low inocula of 1.5 × 10^3^–5.0 × 10^3^ CFU/mL) (e.g., [4]). Instead, the bound amount of potentially cytotoxic significant components that play a critical role could also be explained by the lack of growth of *M. smegmatis* under Visipaque in broth microdilution in a minimal medium. This contains one of the highest iodine concentrations of all tested iodine-containing agents (320 mg iodine per ml compared to 300 mg iodine per ml). Furthermore, the results of the antibacterial efficacy tests in blood culture bottles, as well as those of the agar disk diffusion, showed that a higher concentration of iodine in X-ray contrast media to achieve the described effects is necessary, which has been assumed in previous studies [4,10,13]. With regard to MRI contrast agents (Multihance with 334 mg/mL gadobenic acid and Dotarem with 279 mg/mL gadoteric acid), we found no equal evidence for this. In contrast, the lack of effect of MRI contrast agents on *M. smegmatis* may be related to the fact that this pathogen is also often found in soil samples and, as an environmental microorganism, is better able to deal with heavy metals.

Our other key results focus on the unique findings concerning the effect of pH on contrast media. Only when the pH was reduced from 7.3 to 5.5 could a bacteriostatic effect be shown in *B. subtilis* with Ultravist 370. Likewise, when the pH was reduced from 7.4 to 5.5, inhibition zones were no longer detected not only with Ultravist 370 but also with Telebrix Gastro 300, Multihance and Dotarem. There is no sporadic literature available regarding pH changes and the impact on microorganisms in the presence of contrast agents. So far, only Narins and Chase [6] have reported otherwise, that pH changes do not affect the effect of Hypaque (diatrizoate sodium, drug class Ionic-iodinated contrast media) on microorganisms. Whether the free components or the possibly provoked instability of the complexes with the release of the bound components is responsible for this effect cannot be clearly assigned, and further studies are needed. So far, however, it is assumed that the stability of MRI contrast agents is pH-dependent, but it is unclear how much free gadolinium is or will be present [14]. Our opposite result of Telebrix Gastro 300 and bacteriostatic effect on *P. aeruginosa* at pH 5.5 and bactericidal effect at pH 7 needs an explanation in this context. A possible explanation could be that the higher pH, which leads to a shift of the equilibrium from elemental iodine to hypoiodous acid, consequently acts more effectively against specific bacterial classes such as *P. aeruginosa* [15,16,17,18]. The disinfection effectiveness of different iodine species could already be shown [18,19,20]. Furthermore, the lack of interclass difference (agar disk diffusion at pH 5.5 and broth microdilution in a minimal medium) suggests that the specific properties of the contrast media compositions, such as thermodynamic stability and kinetic inertness (macrocyclic Gd(III) complexes vs. linear and ionic vs. nonionic iodinated contrast media) do not achieve the main effect in abscess collections [21,22].

Another critical aspect is that only about three relevant papers on this topic have been published in the last 20 years. On the one hand, this is because the contrast media most commonly used to date (e.g., Telebrix 1972), including the MRI contrast media (e.g., Dotarem 1989) were developed over several decades towards the end of the last century and the corresponding studies on patient safety had to be published between invention and approval [23,24,25,26]. On the other hand, the importance of image-guided interventions increased with an evident time lag [27]. Consequently, relevant interdisciplinary studies on this specific topic are lacking. Considering that the significance of percutaneous, CT- and MRI-guided interventions with luminal use of contrast medium to visualize abscess collections and possible associated fistula tracts is rapidly gaining over surgical repair, our results are a first step to filling this important gap.

### Limitations

Our limitations are the lack of direct transferability from in vitro to in vivo systems and the lack of testing of anaerobic pathogens, which are also major representatives of abscess collections. For example, it is possible that the altered phagocyte activity is due to the presence of contrast agents; the results in vivo again differ greatly [28]. Furthermore, we did not measure the effective free concentration of the major components (iodine and gadolinium), which will be of great interest to future studies

## 4. Materials and Methods

Six radiographic contrast media were tested in total. Four of them were iodinated contrast media: Ultravist 370 (agent: iopromid, Bayer Schweiz AG, Zurich, Switzerland, monomeric, nonionic, low osmolar, containing 300 mg iodine/mL), Iopamiro 300 (agent: iopamidol, Bracco Suisse SA, Ticino, Switzerland, monomeric, nonionic, low osmolar, containing 300 mg iodine/mL), Telebrix Gastro 300 (agent: meglumin, Guerbet AG, Zurich, Switzerland, monomeric, ionic, high osmolar, containing 300 mg iodine/mL), Visipaque (agent: iodixanol, dimeric, nonionic, iso osmolar, containing 320 mg iodine/mL). Furthermore, the two products containing gadolinium ions were MultiHance 0.5 mmol/mL (agent: gadobenic acid, Bracco Suisse SA, Ticino, Switzerland, containing 334 mg gadobenic acid/mL) and Dotarem (agent: gadoteric acid, Guerbet AG, Zurich, Switzerland containing 279.32 mg gadoteric acid/mL).

All iodinated contrast agents consist of a central element, the tri-iodinated benzene ring. Three iodine atoms covalently bonded to the benzene ring, on the one hand, create a local concentration of iodine, and on the other hand, this organic, functional group reduces the risk of free iodine [21]. The potentially highly reactive and consequently toxic benzene ring is protected from oxidation by side chains [29].

Gd(III) ions are also toxic and consequently bound by chelates or ligands, which are arranged linearly or cyclically [30]. Descriptive statistical analysis was performed after the following tests.

### 4.1. Bacterial Strains

The selected bacterial strains are clinically relevant entities with high loads in abscesses. They are responsible for many abscesses in the clinic. *M. smegmatis* was chosen, as it has a long doubling time, which was helpful in certain experiments. Six different bacterial strains were used in the different experiments of this study: *S. aureus* (ATCC 25923), *P. aeruginosa* (ATCC 27853), *B. subtilis* (DSM 618) and *B. subtilis* (spore suspension, Merck 110649, number of germinable spores 8 × 10^6^ to 5 × 10^7^ CFU/mL), *M. smegmatis* (ATCC 35798) and *E. coli* (ATCC 25922). The following methods were chosen similarly to disinfectant and antimicrobial activity tests (e.g., ISO 22196:2011 and DIN EN 13727:2012). The methods were technically available in the laboratory.

### 4.2. Antibacterial Efficacy Tests in Blood Culture Bottles

Suspensions of fresh overnight cultures of *S. aureus, P. aeruginosa * and *B. subtilis* were prepared (2 different densities of 1.5 × 10^8^ CFU/mL to 5.0 × 10^8^ CFU/mL and 1.5 × 10^3^ CFU/mL to 5.0 × 10^3^ CFU/mL). One milliliter of this suspension was incubated with 9 mL of the nondiluted contrast agent Ultravist, Iopamiro, Telebrix Gastro or Visipaque. Two different pH values of 5.5 und 7.0 were chosen to show possible differences in growth. A pH value of 7.0 is optimal for all used microorganisms, and contrast agents are stable at this pH. On the other hand, a pH value of 5.5 simulates the pH value in abscess cavities, and instabilities of the contrast agents may be possible [31,32,33]. The bacterial suspensions were adjusted drop by drop with 0.5 molar HCL. The resulting pH value therefore varied slightly. These solutions were incubated at 37 °C. After 24 h and 48 h, 5 mL was taken out and incubated in an aerobic blood culture bottle (bact/alert R PF Plus, BioMérieux, Marcy-L’Etoile, France) for 10 to 15 h in an automated system for blood cultures. As a negative control, 1 mL of the bacterial test solution was incubated in sterile 0.9% sodium chloride.

### 4.3. Agar Disk Diffusion

A 2 mL ampule of *B. subtilis* spore suspension was added to agar (Merck 110663) before pouring plates (spore concentration of 6400 CFU to 40,000 CFU/mL). The 2 pH conditions of 5.5 (with 0.5 molar HCL) and 7.4 (with 0.5 molar NaOH) were adjusted in the agar. The bacteria were metabolically active, i.e., there was a variation from well to well depending on growth. Four nonimpregnated paper disks (BioRad, Art. No. 66101) were distributed uniformly onto the plates. Ten microliters of contrast medium (Ultravist, Iopamiro, Telebrix Gastro, Visipaque, Multihance and Dotarem) were pipetted, undiluted, onto paper disks. As positive control, erythromycin (5 mg/L, BioRad, Hercules, CA, USA) was used. All experiments were conducted as duplicates and repeated on two consecutive days.

In a second test series, the bacterial strains *S. aureus* and *P. aeruginosa* were used. Of each microorganism dense, homogenous suspensions were prepared (McFarland standard of 0.5 for *S. aureus* and 1.0 for *P. aeruginosa*). A quantity of 200 µL of this suspension was plated onto Müller–Hinton agar plates. Two pH conditions of 5.5 and 7.4 were adjusted in the agar.

Four nonimpregnated paper disks (BioRad, Art. No. 66101) were distributed uniformly onto the plates. 10 µL of contrast medium were pipetted undiluted onto the paper disks. As a positive control, erythromycin (5 mg/L, BioRad) was used. Blank controls without antibiotics were not included as we know from other experiments that the filter paper disks themselves do not inhibit bacterial growth. This was checked in the daily internal QC of our diagnostic laboratory.

### 4.4. Broth Microdilution in Minimal Medium

Dense, homogenous suspensions of the microorganisms *E.coli*, *S. aureus*, *M. smegmatis* and *F. necrophorum* were prepared in Müller–Hinton Broth. The growth time in Müller–Hinton broth growth was 8–12 h for *E. coli* and *S. aureus* and 24 h for *M. smegmatis*. According to our earlier experiments, these species are in the log growth phase after this time span. Then the cultures were transferred into a minimal medium M9 supplemented with 2% glucose (produced in-house in its own media production unit). The bacteria were cultured in this medium until dense suspensions were reached. Next, 1.5 mL was taken out and pelleted by centrifugation. The pellet was washed with PBS and transferred into minimal medium M9. A quantity of 100 µL of all bacterial strains in the two media were placed onto a microplate and incubated with 100 µL of each contrast agent (Ultravist, Iopamiro, Telebrix Gastro, Visipaque, Multihance and Dotarem). Negative and positive controls were included. The plates were covered by a film and incubated at 37 °C aerobic conditions for *E. coli*, *S. aureus* and *M. smegmatis* and anaerobic conditions for *F. necrophorum*.

## 5. Conclusions

The possible correlations obtained from our new results concerning the antibacterial effect of X-ray as well as MRI contrast agents may have far-reaching consequences. Diagnostics for abscess pathogens in materials with contrast agents should involve molecular analytics, i.e., PCR-based methods in case of inhibited culture growth. Furthermore, so far, the use of X-ray contrast media in body cavities has to be monitored only fluoroscopically as well as the MRI contrast media used in this study are not explicitly prohibited for use in body cavities (e.g., [34,35]). Since the number of minimally invasive, percutaneous interventions is steadily increasing compared to surgery, further research must be carried in this direction. Consequently, there is a possibility that, on the one hand, additional therapeutic options may be gained by luminal administration of contrast agents into abscess cavities and, on the other hand, that the mode of administration of these contrast agents may need to be reevaluated.

## Figures and Tables

**Figure 1 ijms-24-03470-f001:**
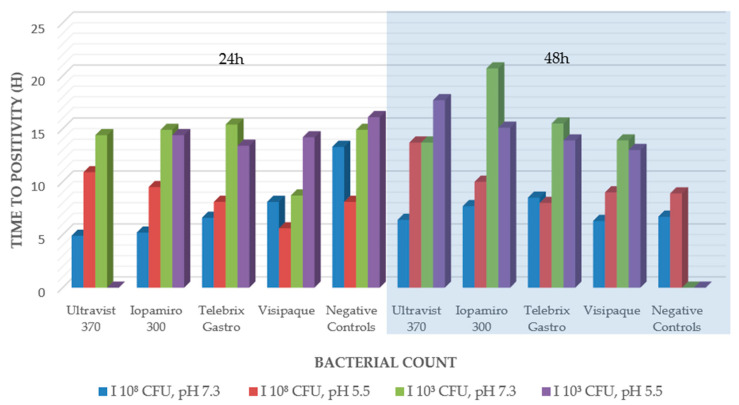
Antibacterial efficacy tests in liquid culture *B. subtilis*.

**Figure 2 ijms-24-03470-f002:**
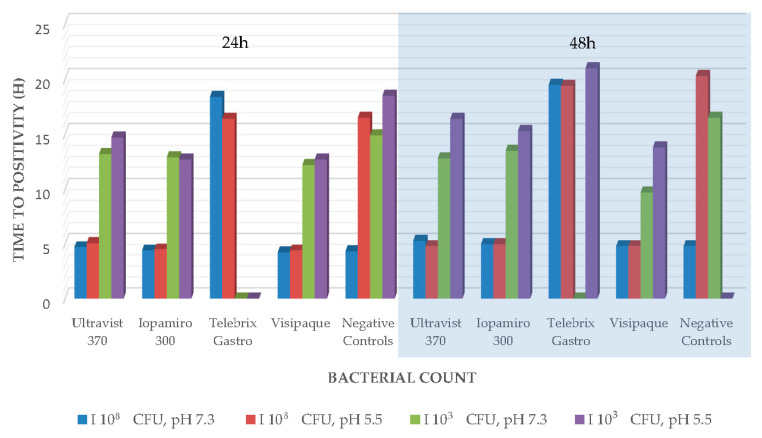
Antibacterial efficacy tests in liquid culture *P. aeruginosa*.

**Table 1 ijms-24-03470-t001:** Indication of the time interval until the growth of the bacteria was detected. Missing growth is highlighted in red color.

			*Staphylococcus* *aureus*		*Bacillus subtilis*		*Pseudomonas aeruginosa*	
Contrast Agent	CFU/mL	pH	24 h	48 h	24 h	48 h	24 h	48 h
Ultravist 370	1.5 × 10^8^–5.0 × 10^8^	7.3	3.8	4.7	4.9	6.4	4.7	5.3
		5.5	4.3	5.8	10.9	13.7	5.1	4.8
	1.5 × 10^3^–5.0 × 10^3^	7.3	12.3	16	14.4	13.7	13.2	12.8
		5.5	11.6	13.7		17.7	14.7	16.4
Iopamiro 300	1.5 × 10^8^–5.0 × 10^8^	7	3.7	4.9	5.2	7.7	4.4	5
		5.5	4.1	5.2	9.5	10	4.5	5
	1.5 × 10^3^–5.0 × 10^3^	7	12.8	14.4	14.9	20.7	12.9	13.5
		5.5	14.9	21.5	14.4	15.1	12.7	15.3
Telebrix Gastro	1.5 × 10^8^–5.0 × 10^8^	7.1	6.6	9.8	6.6	8.5	18.4	19.5
		5.5	5.9	8	8.1	8	16.4	19.4
	1.5 × 10^3^–5.0 × 10^3^	7.1	16.5	21.2	15.4	15.5		
		5.5	11.1	15.9	13.4	13.9		21
Visipaque	1.5 × 10^8^–5.0 × 10^8^	7.2	3.9	6.7	8.1	6.3	4.2	4.8
		5.5	5.4	10.7	5.6	9	4.4	4.8
	1.5 × 10^3^–5.0 × 10^3^	7.2	13.2	20.2	8.7	13.9	12.2	9.7
		5.5	15.5	23.5	14.2	13	12.7	13.8
Negative Controls	1.5 × 10^8^–5.0 × 10^8^	7	7.5	7.8	13.3	6.7	4.3	4.8
		5.5	9.1	18.3	8.1	8.9	16.5	20.3
	1.5 × 10^3^–5.0 × 10^3^	7	23.7		14.9		14.9	16.5
		5.5	16		16.1		18.5	

**Table 2 ijms-24-03470-t002:** Growth of bacteria in minimal medium M9 in the presence of contrast medium.

Bacterium	Ultravist	Iopamiro	Telebrix Gastro	Visipaque	Multihance	Dotarem
*M.smegmatis*	x	x	x	-	x	x
*S.aureus*	-	-	-	-	-	-
*E.coli*	-	x	-	x	x (1/3)	x (1/3)

## Data Availability

Any data or material that support the findings of this study can be made available by the corresponding author upon request.

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
