# Peer review of "Antibacterial Effects of X-ray and MRI Contrast Media: An In Vitro Pilot Study"

_ijms, 2023, doi:10.3390/ijms24043470_

Round 1

Reviewer 1 Report (Previous Reviewer 2)

First congrats on the study, and all the work to make it possible.

I only have 1 remark to make, in the line 157, if you could change, if you agree, or improve:

". After 48 h, the pathogen M. smegmatis showed growth with the following five X-ray contrast media; Ultravist, Iopamiro, Telebrix Gastro, Multihance, and Dotarem (Table 2). The only contrast agent that has not shown growth was Visipaque"

To:

". After 48 h, the pathogen M. smegmatis 155 showed growth with the following five X-ray contrast media; Ultravist, Iopamiro, Teleb- 156 rix Gastro, Multihance, and Dotarem (Table 2), being Visipaque the only contrast agent that has not shown growth."

Its only a minor minor thing. This comment is only because if you take the setence out of order you can think visipaque will work with all bacteria tested.

Kind regards,

Author Response

According to:

. After 48 h, the pathogen M. smegmatis 155 showed growth with the following five X-ray contrast media; Ultravist, Iopamiro, Teleb- 156 rix Gastro, Multihance, and Dotarem (Table 2), being Visipaque the only contrast agent that has not shown growth."

Thank you for your efficient analysis of our manuscript. We fully agree with your proposal and have clearly modified it in line 159/160

Reviewer 2 Report (New Reviewer)

Dear Authors, 

I would like to direct some suggestions to your article:

1. The references are VERY old - only 3 of them are from the past 20 years, the other ones are older, which definately should be changed, because the topic went into advancements. Please, find the papers by Wezgowiec et al., Nimonka et al. concerning the disinfection and sterilization.

2. Line 32 - please, delete the yellow color

3. Line 36, 40 - delete "e.g." from the citation

4. Table 1 - please explain why there are empty "spots" in there?

5. Line 79, 82, 154, 176-180, 200, 255-261, 268-285, 294, 307+ - delete highlightning

6. The materials and methods are well described, although to my mind there should be information why these species of bacteria were chosen, not the others. It should also be stated what is new in this research. It should also contain the statistic section, which is missing with the description of the tests.

7. In the discussion, please add:

- the plasma disinfection as a novel and probably safe method

- the influence of disinfection on the materials used in medicine (eg. silicone in dentistry)

- the use of natural polymers, including chitosan and propolis for disinfection

- the other interesting aspect is that the procedures, such as endodontic treatment, require X-rays, although the additional bacteriostatic and biotic specimens are used to eradicate bacteria, eg. 

Haroon S, Khabeer A, Faridi MA. Light-activated disinfection in endodontics: A comprehensive review. Dent Med Probl. 2021;58(3):411–418. doi:10.17219/dmp/133892

Pelozo LL, Silva-Neto RD, de Oliveira LPB, Salvador SL, Corona SAM, Souza-Gabriel AE. Comparison of the methods of disinfection/sterilization of extracted human roots for research purposes. Dent Med Probl. 2022;59(3):381–387. doi:10.17219/dmp/144762

- note that the effectiveness of decontamination also depends on the structure that is evaluated, which has a high importance in in vitro-in vivo results, and therefore should be dicussed, eg. 

Kubasiewicz-Ross P, Fleischer M, Pitułaj A, Hadzik J, Nawrot-Hadzik I, Bortkiewicz O, Dominiak M, Jurczyszyn K. Evaluation of the three methods of bacterial decontamination on implants with three different surfaces. Adv Clin Exp Med. 2020 Feb;29(2):177-182. doi: 10.17219/acem/112606.

Adding those aspects to the discussion, should also "refresh" your references.

8. Please, add the limitations of the study

After those changes, the paper should be reevaluated.

Round 2

Reviewer 2 Report (New Reviewer)

Dear Authors,

thank you for the explanations.

If you could only find the old references, maybe it would be beneficial to construct figure similar to prisma flow diagram on the search criteria and a short explanation why the references are that old. Try to highlighten that, because it is a very important information in your study.

Also, the limitations should be in a separate chapter. 

Thank you in advance for the corrections. 

Author Response

We have also concluded that your aspect of the old references needs to be better weighted in the paper. Consequently, we have devoted a section to it in the discussion. To do this, we have conducted a new literature search and cited five more papers for providing a coherent explanation. We did not include a prism flow diagram in order not to steer the focus of the paper toward a systematic review. I hope you understand this, you like the additional section in the discussion, and it meets your expectations.

We have also included the limitations in a separate chapter.

Round 3

Reviewer 2 Report (New Reviewer)

Dear Authors,

thank you for understanding my point. I think the changes that you have done are acceptable right now and highlighten that the topic you touched needs further investigations. Best regards

This manuscript is a resubmission of an earlier submission. The following is a list of the peer review reports and author responses from that submission.

Round 1

Reviewer 1 Report

The manuscript ID ijms-2061661 deals with the antibacterial effects of different X-ray and MRI contrast agents. The authors choose 6 contrast agents containing iodine or gadolinium, and tested them facing 6 bacteria species: P. aeruginosa (ATCC 27853), B. subtilis (ATCC ??? And Merck 110649), Mycobacterium smegmatis (ATCC 35798), E. coli (ATCC 25922) and Fusobacterium necrophorum (ATCC 25286). Various tests were used to evaluate the antibacterial activity of these contrast agents: antibacterial efficacy tests in blood culture bottles, agar disk diffusion, and broth microdilution in minimal medium. However, even if a lot of data are presented, there are some lacks in terms of experimental specifications and choices, presentation of results and discussion. 

For these reasons the reviewer recommends that this manuscript be rejected

Below are some of the reviewer’s comments, hoping that they will help the authors to improve their manuscript before submitting it again. 

Bacteria species must be in italic.

Where is the contrast media composition discussed and compared? And the results in regard to this? 

The authors must justify their choices of methods, bacteria strains, etc.

p1 l16, p2 table 1, p3 figure 1&3 and l 79, p6 l156, 158 and 168, and p7 l 209: what are the correct pH values? Why such values, variations, etc.?

p1 l32, p7 l202, 206 and 207: for some numbers superscript format must be used.

p2 l53-54: are the negative controls correct? In other words, how can the results obtained be considered correct?

p2 l 57 & 64: What do “growth after a few hours of incubation” and “low bacterial count” mean? How many? 

p3 l75 to p4 l89: what about the blanck control (i.e. disk without antibiotics)?

p4 l98 and p8 l234: is it glycerine or glucose? 

p4: what’s about control experiments? 

p5 l 146: to which agent media 150 mg corresponds?

p6 l183-192: in this section, the first lines lead readers to believe to that all the contrast agents are iodinated. 

p7 l203: Bacillus subtilis strain should be specified (e.g. ATCC XXX) 

p8 l 226: why did the authors choose 5.5 and 7.4 as pH values?

P8 l 234: What is the supplier of the minimal medium M9 with 2% glucose? 

P8 l232-241: there is no time specification.

Author Response

First, thank you for your precise explanations and the high-quality manuscript analysis. We have taken them very seriously, revised the manuscript in detail over several days, and hope to meet your requirements.

……However, even if a lot of data are presented, there are some

lacks in terms of experimental specifications and choices,

presentation of results and discussion.

Authors' reply: The experimental design corresponds to a small scale pilot study. Therefore, not all immaginable experimental conditions are covered. We think, that the data is noteworthy. It can be expanded in a more comprehensive fullw-up study.

Justification of methods, bacteria strains etc.

The bacterial strains are clinically relevant entities with high case loads in abscesses. The methods were chosen similarly to disinfectant and antimicrobial activity tests (e.g., ISO 22196:2011 and DIN EN 13727:2012). The methods were technically available in the laboratory (added to manuscript, p 8 310-312

p1 l32, p7 l202, 206 and 207: for some numbers superscript and format must be used.

Has been adjusted in the manuscript.

p1 l16, p2 table 1, p3 figure 1&3 and l 79, p6 l156, 158 and 168,

and p7 l 209: what are the correct pH values? Why such values,

variations, etc.?

 the 2 different pH values of 5.5 und 7.0 were chosen to show possible differences in growth. A pH value of 7.0 is optimal for all used microorganisms and contrast agents are stable at this pH. A pH value of 5.5 simulates the pH value in abscess cavities and instabilities of the contrast agents may be possible. This considerations have been added to the main Material&Methods section of the manuscript (p8 319-322).

p2 table 1: For the antibacterial efficacy tests in blood culture bottles the pH of the bacterial suspensions was adjusted drop by drop with 0.5 molar HCL. The resulting pH value therefore varied slightly (added to the manuscript, p8 323). The tables and text state the measured pH.

p3 figure 1&2: approximate pH-value specification

p3 79: Adjustment to pH 7.4 with 0.5 molar NaOH and to pH 5.5 with 0.5 molar HCL. The bacteria were metabiolically active; i.e there is a variation well to well depending on growth (added to the manuscript, p8 333).

p6 156, 158 and 168/ p7 209: variation due to adjustment drop by drop with HCL

p2 53-54: The negative controls with NaCl show no growth after 48 h for the low suspension density. Probably the bacteria died because of the lack of nutritive substances. The lack of a carbon source was not compensated. Still, we consider the controls to be ok since there was growth after 24 h (added to the manuscript, p5 188-190).

p2 57 & 64 Growth after a few hours of incubation means that growth detection was between 3.7 and 9.9 hours for the high bacterial count and between 11.1 and 16.5 for the low bacterial count. Low bacterial count means that 5 ml of a suspension with 102 CFU / ml was put into 10 ml of blood culture liquid (added to the manuscript, p2/3 81-82 and 85).

p3 75 to p4 89: Blank controls without antibiotics were not included as we know from other experiments that the filter paper discs themselves do not inhibit bacterial growth (data not shown). This is checked in the daily internal QC of our diagnostic laboratory (added to the manuscript, p8 346-349).

p4 98 and p8 234: glucose. Its changed on p4 153.

p4: control experiments: Growth controls in M9 supplemented with 2% glucose and M9 of all bacterial strains were included (data not shown in table). (added to manuscript, p4 157)

p5 146: Error is corrected.

P6 183-192: Clarification made (p6 256-266)

p7 203: DSM 618 is corrected, (p7 307)

p8 226: see first answer p1 16

p8 234: The minimal medium M9 supplemented with 2% glucose was made in-house in the own media production unit (added to manuscript, p9 357)

P8 232-241: The growth time in Müller Hinton Broth Growth was 8-12 h for E. coli and S. aureus and 24 h for M. smegmatis. Subsequently the cultures were transferred into minimal medium M9 supplemented with 2% glucose and again growth time was 8-12 h for E. coli and S. aureus and 24 h for M. smegmatis (added to the manuscript, p9 354).

Reviewer 2 Report

Congratulation on the study. 

Regarding the paper, should have been included international standards guideline (NCCLS/CLSI or EUCAST), nerveless your approach seen ok.

The colours on table 1 are not refer in text, and has lack of units. The numbers in "McFarland" should be "CFU/ml"? 

The y axes in in the Figure 1 and 2 should be other name(bacterial count?), Why you present table fo 2 out of 3 bacteria?

Figure 3 should be removed or a lot of improved is needed. There is no explanation in the text or in the figure legend that anyone could understand. There is no uptake of information with this figure.

Figure 4 should be removed as there is not relevant.

Line 232, how you scale being dense and homogenous? by looking? with an densitometer? 

Fusobacterium necrophorum why you added if there is no results? Nor in table 2 or in the text.

You should do a minor revision on the paper and try to add/remove some of the information to present a beter article.

Author Response

First, thank you for your precise explanations and the high-quality manuscript analysis. We have taken them very seriously, revised the manuscript in detail over several days, and hope to meet your requirements.

International standards?

The experiments were done according the published broth microdilution methods. The methods were chosen similarly to disinfectant and antimicrobial activity tests (e.g., ISO 22196:2011 and DIN EN 13727:2012). The methods were technically available in the laboratory (added to manuscript, p 8 319-322.

Colors in table one:

they needed to be wholly correct. Green should show growth, red no growth. We adapt that and refer to the colors in the text. CFU/ml was also added.

Removal of figure 3 is done and figure 4 was improved.

Line 232:

The growth time in Müller Hinton Broth Growth was 8-12 h for E. coli and S. aureus and 24 h for M. smegmatis. According to our earlier experiments, these species are in the log growth phase after this time span. Added to manuscript p9 352-355.

Fusobacterium necrophorum was included into the broth microdilution assay but didn't grow at all, neither in all controls. We removed this microorganism as we don't have any results on it.